# POP-NORM: A THEORETICALLY JUSTIFIED AND MORE ACCELERATED NORMALIZATION APPROACH

## ABSTRACT

Batch Normalization (BatchNorm) has been a default module in modern deep networks due to its effectiveness for accelerating training deep neural networks. It is widely accepted that the great success of BatchNorm is owing to reduction of internal covariate shift (ICS), but recently it is demonstrated that the link between them is fairly weak. The intrinsic reason behind the effectiveness of BatchNorm is still unrevealed that limits it to be made better use. In light of this, we propose a new normalization approach, referred to as Pre-Operation Normalization (POP-Norm), which is theoretically ensured to speed up the training convergence. Not surprisingly, POP-Norm and BatchNorm are largely the same. Hence the similarities can help us to theoretically interpret the root of BatchNorm's effectiveness. There are still some significant distinctions between the two approaches. Just the distinctions make POP-Norm achieve faster convergence rate and better performance than BatchNorm, which are validated in extensive experiments on benchmark datasets: CIFAR10, CIFAR100 and ILSVRC2012.

## 1 INTRODUCTION

The last decade has witnessed significant advances in deep neural networks (DNN), which brought substantial improvements for many real-world tasks, such as object(Szegedy et al., 2015; He et al., 2015), scene(Khan et al., 2016; Guo et al., 2017) and action recognition(Feichtenhofer et al., 2017), object detection(Ren et al., 2015; He et al., 2017b; Redmon & Farhadi, 2018) and image segmentation(Ronneberger et al., 2015; He et al., 2017a). BatchNorm (Ioffe & Szegedy, 2015) is a milestone technique in the development of DNN, and nowadays it has become a default component to construct modern deep networks. BatchNorm is widely-used in DNN due to that it is able to reduce the sensitivity to initialization, significantly raise the learning rate, substantially speed up the training process and considerably improve the performance. Actually, along with this line a variety of works, such as layer normalization (Ba et al., 2016), instance normalization (Ulyanov et al., 2016), weight normalization(Salimans & Kingma, 2016) and group normalization (Wu & He, 2018), batch renormalization, (Ioffe, 2017) were sequently proposed and also have made great success.

Nowadays, the practical success of BatchNorm is mainly attributed to reduction of internal covariate shift (ICS) by controlling the first two moments (mean and variance) of the distributions of layer inputs. Recently, this point of view is challenged by (Santurkar et al., 2018). It points out that the link between reduction of ICS and performance improvement of BatchNorm is tenuous, at best. However, the success of BatchNorm is indisputable, therefore there may exist a profound mathematical mechanism behind BatchNorm. In fact, from the perspective of loss landscape, (Santurkar et al., 2018) demonstrates BatchNorm will make the optimization landscape smoother, and then this smoothness induces a more predictive and stable behavior of the gradients.

As stated above, the original theoretical foundation of the vanilla BatchNorm is not so solid and the exact reason for its effectiveness is poorly understood, hence BatchNorm maybe not optimal in theory and we might not make better use of it. In light of this, we try to propose a theoretically justified normalization approach for accelerating training deep neural networks, so that we can continuously further algorithmic progress. To achieve this goal, we firstly demonstrate that lowering gradient Lipschitz constant and gradient variance is the key to boost the convergence rate of optimization algorithms with stochastic gradient methods. And then we construct a normalization approach – removing the mean of inputs and dividing by the scaled $l_2$ norm of inputs before conducting the

main operation (convolution or inner-production) of a layer, which can be theoretically ensured to reduce the gradient Lipschitz constant and gradient variance.

The new normalization approach and BatchNorm are similar. Hence we utilize their similarity to interpret the effectiveness of the vanilla BatchNorm, along the line of lowering gradient Lipschitz constant and gradient variance, which is different from the view of the loss landscape (Santurkar et al., 2018). According to our theory, we can easily explain why BatchNorm with ReLU is powerful with but ineffective with Sigmoid and Tanh as activation functions, which may help us to better understand the underlying complexity of neural networks.

The main contributions of this paper are summarized as follows:

- From scratch a theoretically justified normalization approach is deduced for accelerating the convergence speed of training deep neural networks.
- We theoretically explain the root of BatchNorm's effectiveness and the weakness from the perspective of lowering the gradient variance and gradient Lipschitz constant.
- With the help of the new normalization approach, in extensive experiments the performance for Sigmoid and Tanh achieve massive improvement, and they are even competitive to ReLU.

Our paper is organised as follows. In Section 2, we will deduce the theoretical premises to accelerate stochastic optimization. Then, built on the premise, we will propose a new normalization approach in Section 3. In Section 4, we will analyze the exact root of BatchNorm's effectiveness. In Section 5, we will implement extensive experiments to demonstrate the superiorly of the new normalization approach. We will summarize our work and discuss further work in Section 6.

## 2 THEORETICAL PREMISES

In this section, we will theoretically deduce that reduction of gradient Lipschitz constant and gradient variance plays a vitally important role in convergence when applying stochastic gradient descent(SGD) algorithms for machine learning tasks.

In a machine learning task, given a set of samples $\{x_i\}_{i=1}^n$ , the empirical risk as the average of all the samples loss is typically treated as the optimizing objective, *i.e.*,

$$F(w) = F\left(w; \{x_i\}_{i=1}^n\right) = \frac{1}{n} \sum_{i=1}^n f_i(w; x_i), \tag{1}$$

where $f_i(w; x_i)$ is the loss incurred by the parameter vector $w$ with respect to the $i$-th sample or mini-batch. Noth that sometimes $F\left(w; \{x_i\}_{i=1}^n\right)$ and $f_i(w; x_i)$ may be abbreviated as $F(w)$ and $f_i(w)$.

In large-scale machine learning tasks, such as deep learning, we usually choose SGD to minimize the empirical risk in Eq. (1) since its computational cost is cheap. Applying SGD, the parameter update at $k$-th iteration is

$$w_{k+1} \leftarrow w_k - \alpha_{k+1} \nabla f_{i_k}(w_k; x_{i_k}) \tag{2}$$

where at the $k$-th iteration $x_{i_k}$ is *randomly* chosen from the full sets $\{x_i\}_{i=1}^n$, and $\alpha_{k+1}$ is the learning rate.

Convergence guarantee of most gradient-based algorithms including SGD is built upon the assumption of Lipschitz-continuous objective gradients, which is state as follow.

The function $F(w)$ is continuously differentiable and the gradient function of $F(w)$, namely $\nabla F(w)$, is Lipschitz continuous with gradient Lipschitz constant $L > 0$, i.e.,

$$\|\nabla F(w_1) - \nabla F(w_2)\| \leq L\|w_1 - w_2\|. \tag{3}$$

An important equality directly induced from Eq. (3) is

$$F(w_2) \leq F(w_1) + \langle \nabla F(w_1), w_2 - w_1 \rangle + \frac{L}{2}\|w_2 - w_1\|^2. \tag{4}$$

The proof for it can be found in (Nesterov, 2004; Bottou et al., 2016).

Under the assumption in Eq. (4), we provide convergence analysis in the following for the general objective function Eq. (1) when applying SGD, no matter whether it is convex or not.

**Theorem 1** *Suppose the assumption in Eq. (4) is satisfied and the gradient of $F(w_k)$ in Eq. (1) is bounded (i.e., $\nabla F(w_k) \leq M \leq \infty$), SGD algorithm is applied to optimize $F(w_k)$.*

*(1) If the leaning rate satisfies $\sum_{k=1}^{\infty} \alpha_k = \infty$ and $\sum_{k=1}^{\infty} \alpha_k^2 < \infty$, and then*

$$\frac{\sum_{k=0}^{K} \alpha_{k+1} \mathbb{E}[\|\nabla F(w_k)\|^2]}{\sum_{k=0}^{K} \alpha_{k+1}} \leq \frac{\mathbb{E}[F(w_0)] - F_{inf}}{\sum_{k=0}^{K} \alpha_{k+1}} + \frac{\frac{L}{2} \sum_{k=0}^{K} \alpha_{k+1}^2 \mathbb{E}[\|\nabla f_{i_k}(w_k)\|^2]}{\sum_{k=0}^{K} \alpha_{k+1}} \xrightarrow{K \to \infty} 0,$$
(5)

*where $F_{inf} = \min_w F(w)$. Then, $\|\nabla F(w_K)\| \xrightarrow{K \to \infty} 0$ in probability .*

*(2) If the leaning rate satisfies $\frac{1}{\alpha_k} \geq L$ and $\sum_{k=1}^{\infty} \alpha_k = \infty$, we have*

$$\frac{\sum_{k=0}^{K} \alpha_{k+1} \mathbb{E}[\|\nabla F(w_k)\|^2]}{\sum_{k=0}^{K} \alpha_{k+1}} \leq \frac{\mathbb{E}[F(w_0)] - F_{inf}}{\frac{1}{2} \sum_{k=0}^{K} \alpha_{k+1}} + \frac{\sum_{k=0}^{K} \alpha_{k+1} \mathbb{V}[\nabla f_{i_k}(w_k)]}{\sum_{k=0}^{K} \alpha_{k+1}},$$
(6)

*where $\mathbb{V}[\nabla f_{i_k}(w_k)] = \mathbb{E}[\|\nabla f_{i_k}(w_k) - \mathbb{E}[\nabla f_{i_k}(w_k)]\|^2]$ that is the gradient variance of $f_{i_k}(w_k)$. When $K \to \infty$,*

$$\inf_k \left( \mathbb{E}[\|\nabla F(w_k)\|^2] \right) \leq \sup_k \left( \mathbb{V}[\nabla f_{i_k}(w_k)] \right).$$
(7)

According to the first conclusion of Theorem 1, when the learning rate satisfies $\sum_{k=1}^{\infty} \alpha_k = \infty$ and $\sum_{k=1}^{\infty} \alpha_k^2 < \infty$, we know that SGD algorithm will guarantee the empirical risk function in Eq. (1) converges to a stationary point in probability. The conditions in the second conclusion of Theorem 1 is easier to be met. In this case we should make the the gradient variance $\mathbb{V}[\nabla f_{i_k}(w_k)]$ as smallest as possible to ensure the empirical risk function in Eq. (1) can converge to a good point.

Theorem 1 also implies that increasing learning rate will directly speed up convergence. Since larger $\alpha_k$ will make the term $\frac{\mathbb{E}[F(w_0)] - F_{inf}}{\sum_{k=0}^{K} \alpha_{k+1}}$ in Eq. (5) (or $\frac{\mathbb{E}[F(w_0)] - F_{inf}}{\frac{1}{2} \sum_{k=0}^{K} \alpha_{k+1}}$ in Eq. (6)) faster to diminish. However, when the learning rate $\alpha_k$ becomes large, the term $\frac{L}{2} \sum_{k=1}^{K} \alpha_k^2 \mathbb{E}[\|\nabla f_{i_k}(w_k)\|^2]$ in Eq. (5) (or $\sum_{k=0}^{K} \alpha_{k+1} \mathbb{V}[\nabla f_{i_k}(w_k)]$ in Eq. (6) ) may also become large and the condition $\frac{1}{\alpha_{k+1}} \geq L$ may not be met. Hence reducing the gradient variance $\mathbb{V}[\nabla f_{i_k}(w_k)]$ [1]and the value of Lipschitz constant $L$ is the key to increase the learning rate and accelerate the convergence. It is worth noting that the conclusion that gradient variance reduction is beneficial to accelerate convergence has been demonstrated in (Johnson & Zhang, 2013) when the objective function is convex.

Therefore, if an normalization approach is effective to make Eq. (1) converge to a good point and accelerate the convergence process, it needs to *reduce gradient variance $\mathbb{V}[\nabla f_{i_k}(w_k)]$ and gradient Lipschitz constant $L$.*

## 3 THE PROPOSED APPROACH

In this section, we will present a new normalization approach of which each step will be theoretically proved to be beneficial to reduce the gradient variance $\mathbb{V}[\nabla f_{i_k}(w_k)]$ and the gradient Lipschitz constant $L$ simultaneously, therefore the new approach will accelerate the convergence of training and improve the performance.

---

[1]It is known $\mathbb{E}[\|\nabla f_{i_k}(w_k)\|^2] = \mathbb{V}[\nabla f_{i_k}(w_k)] + (\mathbb{E}[\|\nabla f_{i_k}(w_k)\|])^2$, decreasing $\mathbb{V}[\nabla f_{i_k}(w_k)]$ will also lower $\mathbb{E}[\|\nabla f_{i_k}(w_k)\|^2]$.

When applying mini-batch SGD, at the $k$-th iteration the empirical risk loss with respect to the weight/bias of the $l$-th fully-connected layer of DNN can be expressed as [2],

$$F_{(m)}(w_k^{(l)}) = \frac{1}{m}\sum_{i=1}^{m} f_{i_k}(w_k^{(l)}; x_{i_k}) = \frac{1}{m}\sum_{i=1}^{m} g\left(\left(x_{i_k}^{(l-1)}\right)^t w_k^{(l)} + b_k^{(l)}\right) = G_{(m)}\left(\left(X_{(m)_k}^{(l-1)}\right)^t w_k^l + B_{(m)_k}^{(l)}\right)$$

(8)

where $m$ is the size of mini-batch, and $X_{(m)_k}^{(l-1)} = [x_{1_k}^{(l-1)}; x_{2_k}^{(l-1)}; ...; x_{m_k}^{(l-1)}]$ and $B_{(m)_k}^{(l)} = [b_{1_k}^{(l)}, b_{2_k}^{(l)}, ..., b_{m_k}^{(l)}]$. $x_{i_k}^{(l-1)}$ is randomly choose from the entire train set $\{x_{i_k}^{(l-1)}\}_{i=1}^{n}$ with e-qual probability, hence $F(w_k^{(l)}) = \frac{1}{n}\sum_{i=1}^{n} f_{i_k}(w_k^{(l)}; x_{i_k}^{(l-1)})$ is the expectation of $F_{(m)}(w_k^l)$( $\mathbb{E}[F_{(m)}(w_k^l)] = F(w_k^l)$), where $n$ is the total number of train set. Note that Eq. (8) also implies that $F(w_k^l) = G\left((X_k^{l-1})^t w_k^l + B_k^l\right)$ where $X_k^{l-1} = [x_{1_k}^{l-1}; x_{2_k}^{l-1}; ...; x_{n_k}^{l-1}]$ and $B_k^l = [b_{1_k}^l, b_{2_k}^l, ..., b_{n_k}^l]$. For clarity, henceforth we will omit the layer index.

We construct a new normalization approach, referred to as Pre-Operation Normalization (POP-Norm), of which one main characteristic is putting the normalization technique before operation. For a layer fed with a $d$-dimensional input $x_{i_k} = (x_{i_k}^{[1]}, x_{i_k}^{[2]}..., x_{i_k}^{[d]})$ , we ideally normalize each dimension with the global estimation of entire samples, *i.e.*,

$$\bar{x}_{i_k}^{[j]} = x_{i_k}^{[j]} - \eta^{[j]} \cdot \frac{1}{n}\sum_{i=1}^{n} x_{i_k}^{[j]}, \qquad \text{(Mean Removal)} \qquad (9)$$

$$\hat{x}_{i_k}^{[j]} = \frac{\bar{x}_{i_k}^{[j]}}{\rho^{[j]} \cdot \sqrt{\frac{1}{n}\sum_{i=1}^{n}\left(\bar{x}_{i_k}^{[j]}\right)^2}} = \frac{\bar{x}_{i_k}^{[j]}}{\frac{\rho^{[j]}}{\sqrt{n}}\|x_k^{[j]}\|_2}, \qquad \text{(Division by sclaed } l_2 \text{ Norm)} \qquad (10)$$

where $0 \le \eta^{[j]} \le 2$, $\rho^{[j]}\sqrt{\frac{1}{n}\sum_{i=1}^{n}\left(\bar{x}_{i_k}^{[j]}\right)^2} \ge 1$ and $x_k^{[j]} = (x_{1_k}^{[j]}, x_{2_k}^{[j]}, ..., x_{n_k}^{[j]})$.

The normalization approach consists of two parts – mean removal and division by $l_2$ norm. The both steps are actually not interdependent and can individually take effects for decreasing gradient variance and gradient Lipschitz constant. We will theoretically demonstrate it in the following.

**Theorem 2 (Justification of Mean Removal in Eq.(9))** *Eq. (9) is adopted to handle each input $x_{i_k}$ of the model in Eq. (8). Suppose assumption in Eq. (3) is satisfied and the gradient and Hessian function of $g(z)$ and $G(z)$ in Eq. (8) is bounded , i.e., $\|\nabla g(z)\|_2 \le M_1 < \infty$ and $\|\nabla^2 G(z)\|_2 \le M_2 < \infty$ , we have*

*(1) the upper bound of the gradient variance $\mathbb{V}[\nabla f_{i_k}(w_k; \bar{x}_{i_k})]$ will be lower than that of $\mathbb{V}[\nabla f_{i_k}(w_k; x_{i_k})]$.*

*(2) the upper bound of the minimal gradient Lipschitz constant of $F\left(w_k; \{\bar{x}_{i_k}\}_{i=1}^n\right)$ will be lower than that of $F\left(w_k; \{x_{i_k}\}_{i=1}^n\right)$;*

**Theorem 3 (Justification of Division by $l_2$ Norm in Eq.(10))** *Eq. (10) is adopted to handle each input $\hat{x}_{i_k}$ from Eq. (9). Suppose assumption in Eq. (3) is satisfied and the gradient and Hessian function of $g(z)$ and $G(z)$ in Eq. (8) is bounded , i.e., $\|\nabla g(z)\|_2 \le M_1 < \infty$ and $\|\nabla^2 G(z)\|_2 \le M_2 < \infty$ , we have*

*(1) the upper bound of the gradient variance $\mathbb{V}[\nabla f_{i_k}(w_k; \hat{x}_{i_k})]$ will be lower than that of $\mathbb{V}[\nabla f_{i_k}(w_k; \bar{x}_{i_k})]$ ;*

*(2) the upper bound of the minimal gradient Lipschitz constant of $F\left(w_k; \{\hat{x}_{i_k}\}_{i=1}^n\right)$ will be lower than that of $F\left(w_k; \{\bar{x}_{i_k}\}_{i=1}^n\right)$;*

*(3) if $F(w_k)$ is locally convex and $\frac{\max_j((\rho^{[j]})^2)}{(\min_j(\rho^{[j]})^2)} \le \frac{\max_j\left(\sum_{i=1}^n(\bar{x}_{i_k}^{[j]})^2\right)}{\min_j\left(\sum_{i=1}^n(\bar{x}_{i_k}^{[j]})^2\right)}$, the lower bound of conditional number of $\nabla^2 F\left(w_k; \{\hat{x}_{i_k}\}_{i=1}^n\right)$ will be lower than that of $\nabla^2 F\left(w_k; \{\bar{x}_{i_k}\}_{i=1}^n\right)$ in the local space.*

---

[2] A convolution layer can be also reformulated to be a matrix form , hence the analysis for convolution layers is similar, but it is still somewhat complicated and we discuss it in the appendix.

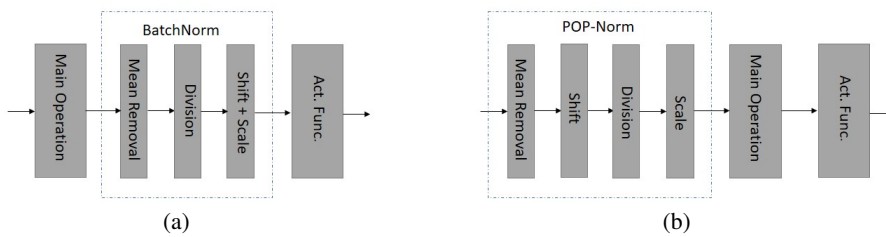

Figure 1: (a) The structure of a standard layer with BatchNorm. (b) The structure of a layer with POP-Norm.

Due to the limited space, the proofs of Theorem 2 and Theorem 3 are provided in the appendix.

The computation of $\frac{1}{n}\sum_{i=1}^{n} x_{i_k}$ and $\frac{1}{n}\sum_{i=1}^{n} \bar{x}_{i_k}^2$ in Eq.(9 - 10) over the entire training set is impractical when adopting SGD. Therefore, just like the vanilla BatchNorm, each mini-batch produces $\frac{1}{m}\sum_{i=1}^{n} x_{i_k}$ and $\frac{1}{m}\sum_{i=1}^{n} \bar{x}_{i_k}^2$ as the estimates instead. This implementation will also make all the inputs fully participate in the gradient back-propagation. The vanilla BatchNorm introduces the scale and shift parameter to be learned to restore the presentation power of the network. To make POP-Norm more like the vanilla BatchNorm, we factorize $\eta^{[j]}$ and $\rho^{[j]}$ into a fixed parameter and a parameter to be learned. In practice POP-Norm is presented as follows:

$$\bar{x}_{i_k}^{[j]} = x_{i_k}^{[j]} - \frac{1}{m}\sum_{i=1}^{m} x_{i_k}^{[j]} + \beta^{[j]}, \qquad \text{(Mean Removal)} \tag{11}$$

$$\hat{x}_{i_k}^{[j]} = \frac{\gamma^{[j]}}{\kappa} \cdot \frac{\bar{x}_{i_k}^{[j]}}{\sqrt{\frac{1}{m}\sum_{i=1}^{m}\left(\bar{x}_{i_k}^{[j]}\right)^2}}, \qquad \text{(Division by scaled $l_2$ Norm)} \tag{12}$$

where $\beta^{[j]}$ and $\gamma^{[j]}$ are the shift parameter and the scale parameter to be learned, and $\kappa$ ($\geq 1$) is a hyperparameter. If $|\beta^{[j]}| < \frac{1}{m}\sum_{i=1}^{m} x_{i_k}^{[j]}$ and $|\gamma^{[j]}| < \kappa\sqrt{\frac{1}{m}\sum_{i=1}^{m}\left(\bar{x}_{i_k}^{[j]}\right)^2}$ are satisfied, Theorem 2 and Theorem 3 will still hold.

**Remark 1.** Although the main steps of POP-Norm in Eq.(11 - 12) are similar to the vanilla BatchNorm, they are still slight different, as shown in Figure 1. For BatchNorm, $\hat{x}_{i_k}^{[j]} = \gamma^{[j]}\left(x_{i_k}^{[j]} - \mu^{[j]} + \beta^{[j]}\right)/\sqrt{\frac{1}{m}\sum_{i=1}^{m}\left(x_{i_k}^{[j]} - \mu^{[j]}\right)^2}$ where $\mu = \frac{1}{m}\sum_{i=1}^{m} x_{i_k}^{[j]}$, while for POP-Norm, $\hat{x}_{i_k}^{[j]} = \gamma^{[j]}\left(x_{i_k}^{[j]} - \mu^{[j]} + \beta^{[j]}\right)/\sqrt{\frac{1}{m}\sum_{i=1}^{m}\left(x_{i_k}^{[j]} - \mu^{[j]} + \beta^{[j]}\right)^2}$. Besides, there are other two significant differences between the proposed POP-Norm and the vanilla BatchNorm – placing normalization before the main operation(convolution and inner-production) and adding a new scale parameter $\kappa$. Actually, we deliberately make POP-Norm more like the the vanilla BatchNorm. POP-Norm has other forms, for example, mean removal in Eq.(11) can be $\bar{x}_{i_k}^{[j]} = x_{i_k}^{[j]} - \frac{1+\beta^{[j]}}{m}\sum_{i=1}^{m} x_{i_k}^{[j]}$, and the $l_2$ norm to divide can be substituted by the other norm, such as $l_1$ norm and $l_\infty$ nrom. Mean removal and division by $l_2$ norm of POP-Norm are actually not interdependent and can individually take effects for decreasing gradient variance and gradient Lipschitz constant, and their order can be also exchanged. From the proof of Theorem 2 and Theorem 3, we also know the division by $l_2$ norm will lowering the upper bound of $\mathbb{V}[\nabla f_{i_k}(w_k)]$ and $L$ more substantially than mean removal. Additionally, the theoretical analysis and construction of POP-Norm are generalized, and we can also apply them to construct a new normalization approach to compete with instance normalization (Ulyanov et al., 2016) and group normalization (Wu & He, 2018).

**Remark 2.** Adding a new parameter $\kappa$ and setting $\kappa \geq 1$ is more beneficial to decrease the gradient variance and the gradient variance $\mathbb{V}[\nabla f_{i_k}(w_k)]$ and the gradient Lipschitz constant $L$. When looking into the proof of Theorem 2 and Theorem 3, we can conclude the upper bound of $\mathbb{V}[\nabla f_{i_k}(w_k)]$ and $L$ will be proportional to $\frac{1}{\kappa^2}$, which means the larger $\kappa$ will be quickly smaller $\mathbb{V}[\nabla f_{i_k}(w_k)]$ and $L$ so that speeds up the convergence. However, $\kappa$ can not be too large, otherwise the value after the operation will be much small, and then it may harm the representation capability of the layer,

since the non-linear activation function will mainly work in the linear regime. Moreover, when $\kappa$ is too large that will directly make gradient variance much small, and then it is difficult for train loss to escape the trap of early local minimums. In our practices, when $\kappa$ is moderately large, it will substantially improve the performance, which is demonstrated in the experimental section. Not strictly speaking, adding $\kappa$ is just make $f(w)$ change to $f(\frac{w}{\kappa})$. It seems the optimization trajectory will be the same if we magnify the initial value of $w$ by an exact factor $\kappa$ and scale the corresponding learning rate. However, we do not exactly change the initial value and the learning rate, and the optimum is not unique. On the contrary, the number of local minimums of a deep neural network is much large and they scatter throughout all the space. Therefore, adding $\kappa$ is more likely to help $w$ in the network quickly converge to a local minimum near the initial value according to Theorem 3 rather than along with a far trajectory converge to the original local minimum.

**Remark 3.** As proved in Theorem 3, adopting the POP-Norm will be conducive to reduction of the condition number of $\nabla^2 F(w_k)$ if $F(w_k)$ is locally convex. Smaller condition number will make the training algorithm avoid the danger of running into a sudden change of the loss such as a flat region or a steep slope, which also enable us to adopt a larger learning rate. In other words, the smaller condition number will make the landscape or convergence trajectory of the loss is more smooth and substantially speed up the training process.

## 4 JUSTIFYING EFFECTIVENESS AND WEAKNESS OF BATCHNORM

The vanilla BatchNorm has been a standard toolkit for modern deep networks, but the root of Batch-Norm's effectiveness is still vague. The motivation of BatchNorm built on ICS is somewhat heuristic. Actually, in (Santurkar et al., 2018) it demonstrates the link between the performance gain and reduction of ICS is fairly weak, which means the success of BatchNorm may be not owing to ICS. (Santurkar et al., 2018) owes faster training with BatchNorm to the improvement of loss landscape. In this section, we will analyze not only BatchNorm's effectiveness but also its weakness from the perspective of lowering the gradient variance and gradient Lipschitz constant above.

BatchNorm and POP-Norm are similar, but there are still some significant distinctions between BatchNorm and POPNorm. One of them is putting the normalization before the main operation (convolution and inner-production) for POPNorm. As analyzed in the above subsection, each step of POP-Norm has been theoretically justified. Therefore, we can analyze the influence of differences between BatchNorm and POPNorm to verify the effectiveness and weakness of BatchNorm.

BatchNorm is placed behind the operation of convolution or inner-production and before the non-linear activation function which is displayed in Figure 1, hence BatchNorm actually takes effect on the operation of the next layer. For simplicity, we omit the shift and scale parameter here. Batch-Norm and POP-norm can be expressed as:

$$\bar{x}_{i_k}^{[j]} = x_{i_k}^{[j]} - \frac{1}{n} \sum_{i=1}^{n} x_{i_k}^{[j]} \tag{13}$$

$$\hat{x}_{i_k}^{[j]} = \frac{\bar{x}_{i_k}^{[j]}}{\sqrt{\frac{1}{n} \sum_{i=1}^{n} \left(\bar{x}_{i_k}^{[j]}\right)^2}} \tag{14}$$

$$\tilde{x}_{i_k}^{[j]} = h(\hat{x}_{i_k}^{[j]}) \tag{15}$$

where $h(\cdot)$ is a activation function.

Due to nonlinearity of the activation function, the benefits of normalization will be impaired. When it is ReLU, the damage will be not so serious. Suppose all samples $\{\bar{x}_{i_k}^{[j]}\}_{i=1}^{n}$ are symmetric about zero that can be easy to be satisfied in practice, we have

$$\tilde{x}_{i_k}^{[j]} = h\left(\bar{x}_{i_k}^{[j]} / \sqrt{\frac{1}{n} \sum_{i=1}^{n} \left(\bar{x}_{i_k}^{[j]}\right)^2}\right) = h(\bar{x}_{i_k}^{[j]}) / \sqrt{\frac{2}{n} \sum_{i=1}^{n} h^2\left(\bar{x}_{i_k}^{[j]}\right)}, \tag{16}$$

hence when adopting ReLU as the activation function for BatchNorm, it is equal to taking the step division by $l_2$ norm with respect to $h(\bar{x}_{i_k}^{[j]})$ for POP-Norm. In other words, no matter we put normalization before or behind ReLU, division by $l_2$ norm will be effective. From Remark 2, we know

in normalization mean removal and division by $l_2$ norm can take effects individually and division by $l_2$ norm plays a major role and can still substantially reduce the gradient variance and gradient Lipschitz constant. Therefore, *the effectivness of BatchNorm with ReLU will be still guaranteed*. On the contrary, when the activation function is Tanh, it will damage division by $l_2$ norm and reserve mean removal, but *the performance of BatchNorm with Tanh will be still undermined*. When the activation function is Sigmoid, it will destroy both of mean removal and division by $l_2$ norm, hence *performance of BatchNorm with Sigmoid will be seriously deteriorated*.

As we analyze in Remark 2, moderately increase the scale of the $l_2$ norm will be beneficial to decrease the gradient variance and gradient Lipschitz constant. It is known most of activation functions will shrink the value of inputs, and then the value of normalization will become relatively larger, hence from this perspective of view, activation function may be beneficial, but the deterioration brought by distortion of activation functions may be still dominated for BatchNorm with Sigmoid, but *BatchNorm with ReLU will be benefited from the declined value of inputs*.

## 5 EXPERIMENTS

### 5.1 GENERAL SETTINGS

In the experiments, we first modify each layer in Figure 1(a) to Figure 1(b) for widely-used VGG-nets (Karen Simonyan, 2016) and ResNets (He et al., 2016) with POP-Norm, and all other network architecture and parameters were exactly the same with the standard VGG-nets and ResNets, and then we evaluate the standard VGG-nets and ResNets and the new ones on benchmark datasets: CIFRR10, CIFAR100 and ILSVRC2012. It is worth noting that we just want to evaluate the performance and effectiveness of the proposed POPNorm, compared with BatchNorm, rather than pursue the highest accuracy on benchmark datasets, hence we do not adopt much deep networks and complicated tricks.

For the experiments on the datasets CIFAR10 and CIFAR100, the batch size is set as $128$. We use a weight of $0.0005$ and SGD with a momentum of $0.9$. To simplify the tuning process and compare fairly, we identically start with a learning rate of $0.1$, divided by $10$ at $16k$ and $24k$ iterations, and finally terminate at $30k$ iterations. For the experiments on ILSVRC2012, standard SGD with weight decay of $0.0001$ and momentum of $0.9$ is adopted. the number of samples in a batch size is set as $256$ uniformly distributed on $8$ GPUs. The learning rate starts from $0.05$ and is divided by $10$ at $250k$ and $450k$ and $550k$ iterations, and finally terminate at $600k$ iterations. The images are first simply resized to $256 \times 256$, and then randomly cropped to $224 \times 224$. No other data augmentation except simple horizontal image flipping is employed.

### 5.2 PARAMETER SENSITIVITY ANALYSIS

We add a scaling factor $\kappa$ to magnify the value of normalization for POP-Norm. $\kappa$ of POP-Norm will greatly influence the effectiveness of POP-Norm. In this subsection, we will evaluate the performance of VGG16 Embedding POP-Norm with different values of scaling factor $\kappa$ from the candidate set of $\{1, 4, 16\}$.

As shown in Figure 2(a-c), when $\kappa$ is moderately large ($\kappa = 4$), it is not only beneficial to speed up the training but also help train loss to finally converge to a smaller value. However, if we further increase $\kappa$ to 16, although the relative convergence speed is accelerated, the training loss is more like to converge to a larger local minimum. The reason for it is just what we analyzed in Remark 2 of Section 3. Overlarge $\kappa$ will make activation functions (Tanh and Sigmoid), work in the linear regime, which will further harm the representative ability of neural networks, so that it is difficult for training loss to find a better local minimum. Overlarge $\kappa$ will directly lead to small gradient variance, which will make train loss hard to escape the trap of an early local minimum. $\kappa$ in POP-Norm will be uniformly set to $4$ in the following experiments for better performance.

POP-Norm speed up the convergence through reducing gradient Lipschitz constant (increasing learning rate) and lowering gradient variance. We fix the learning rate in the experiments in this subsection, hence it also experimentally validates reduction of gradient variance is conducive to faster training convergence.

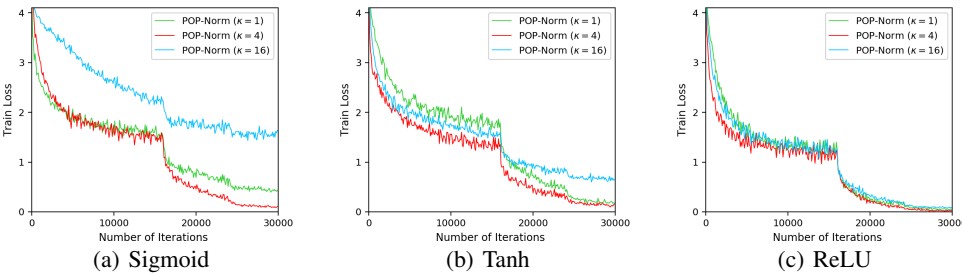

Figure 2: Train Loss for POP-Norm embedded VGG16 with different activation functions on CIFAR100: (a) Sigmoid, (b) Tanh and (c) ReLU.

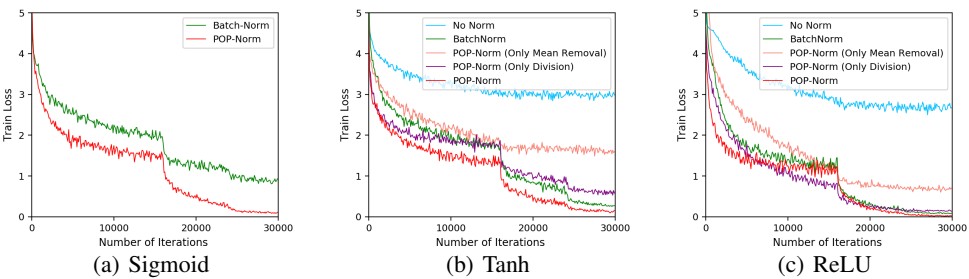

Figure 3: Train Loss for different normalization approaches embedded VGG16 with different activation functions on CIFAR100: (a) Sigmoid, (b) Tanh and (c) ReLU.

## 5.3 ABLATION EXPERIMENTS

As analyzed in Remark 1 in Section 3, mean removal and division by $l_2$-norm of POP-Norm can work individually, in this section we will evaluate their performance of VGG16 on CIFAR100 individually and collectively. To better access their performance, we also add VGG with no normalization and BatchNorm as anchors. We set the initial learning rate 0.00005 and 0.005 for no normalization and POP-norm only with mean removal since a higher learning rate will make training diverge. Specifically, when the activation function is Sigmoid, no matter what value the initial learning rate is, no normalization, POP-norm only with mean removal and POP-norm only with division cannot make loss decrease. This phenomenon for no normalization was also reported in (Ioffe & Szegedy, 2015). We guess this may be owing to gradient vanishing, and the mean removal and division together can relief it but each individual component can not. We will explore the exact reason in the future.

As shown in Figure 3, comparing with no normalization, mean removal and division by $l_2$-norm works can individually accelerate convergence mainly due to increasing learning rate, and it also implies that division by $l_2$-norm is more influential than mean removal in POP-Norm, which can validate our analysis in Section 3. Moreover, when adopting ReLU as activation function, the performance of BatchNorm is close to POP-Norm only with division. And when adopting Tanh and Sigmoid, the convergence rate of BatchNorm is much slower than POP-Norm. These phenomenons just follow the analysis in Section 4.

## 5.4 CLASSIFICATION COMPARISON

In this subsection, we will access the classification performance of BatchNorm and POP-Norm on widely-used CIFAR10, CIFAR100 and ILSVRC2012. The results are reported in Table 1-2.

As displayed in Table 1, the classification accuracy for POP-Norm is consistently higher ( average more than 3%) than BatchNorm. Specifically, when the activation functions are Sigmoid and Tanh,

the improvement of POP-Norm is more obvious, gaining about $8\%$ and $4\%$ respectively, which demonstrates the effectiveness of POP-Norm. It is worth noting that with the help of the proposed POPNorm the performance of the networks with Sigmoid and Tanh is fairly close to the networks with ReLU, which also validates the power of the proposed approach.

Comparing to BatchNorm with ReLU, the performance for BatchNorm with Sigmoid and Tanh is heavily discounted. The reason for it is what we analyzed in Section 4. When the activation function is ReLU, placing BatchNorm behind convolution and inner-production and before activation functions will less influence the performance of normalization, but when adopting Sigmoid or Tanh, this position will have a great impact for the performance of normalization. Therefore, from another aspect, it verifies the importance of putting normalization before convolution and inner-production.

Table 1: Mean test accuracy(%) and standard deviation of 10 trials for BatchNorm and POPNorm embedded VGG16 and ResNet20 on CIFAR10 and CIFAR100.

| Model | Act. Fun. | CIFAR10 | | CIFAR100 | |
|---|---|---|---|---|---|
| | | BatchNorm | POP-Norm | BatchNorm | POP-Norm |
| VGG16 | Sigmoid | $84.21 \pm 0.46$ | $\mathbf{90.69 \pm 0.15}$ | $62.49 \pm 0.56$ | $\mathbf{70.29 \pm 0.14}$ |
| | Tanh | $88.69 \pm 0.15$ | $\mathbf{90.63 \pm 0.09}$ | $66.63 \pm 0.12$ | $\mathbf{70.59 \pm 0.09}$ |
| | ReLU | $90.89 \pm 0.13$ | $\mathbf{92.15 \pm 0.11}$ | $70.83 \pm 0.25$ | $\mathbf{71.38 \pm 0.12}$ |
| ResNet20 | Sigmoid | $83.08 \pm 0.14$ | $\mathbf{89.68 \pm 0.14}$ | $54.10 \pm 0.37$ | $\mathbf{66.91 \pm 0.32}$ |
| | Tanh | $88.08 \pm 0.11$ | $\mathbf{89.63 \pm 0.09}$ | $62.42 \pm 0.36$ | $\mathbf{67.02 \pm 0.10}$ |
| | ReLU | $90.23 \pm 0.13$ | $\mathbf{90.90 \pm 0.12}$ | $66.70 \pm 0.16$ | $\mathbf{67.53 \pm 0.12}$ |

We also implement classification experiments on ILSVRC2012. As shown in Tabel 2, POP-Norm achieves about $0.8\%$ higher classification accuracy than BatchNorm, which demonstrates POP-Norm is also effective in large-scale datasets. Due to the limited time, we did not utilize deeper networks to evaluate BatchNorm and POP-Norm. We will report more results of deeper networks in future.

Table 2: Test accuracy(%) for BatchNorm and POP-Norm embedded ResNet18 on ILSVRC2012. The activation function is ReLU.

| Model | BatchNorm | | POP-Norm | |
|---|---|---|---|---|
| | top-1 | top-5 | top-1 | top-5 |
| ResNet18 | 66.81 | 87.32 | 67.53 | 87.95 |

## 6 CONCLUSIONS AND DISCUSSIONS

In this paper we try to propose a theoretically justified and effective normalization approach. To achieve this goal, we first deduce that lowing gradient variance and gradient Lipschitz constant is the key to speed up the convergence when adopting SGD. And then we present a new normalization approach, of which each step is theoretically proven to be beneficial to reduce gradient variance and gradient Lipschitz constant. POP-Norm and BatchNorm are largely the same, which just help us to explain the effectiveness of BatchNorm. Besides the form of POP-Norm is slightly different from BatchNorm, there are two other differences between the two normalization methods. One difference is that the proposed method place the normalization ahead of the main operation of a layer rather than behind it. Another is that the proposed method adds a scaling factor to increase the value of normalization. Just the differences make our normalization approach more powerful, which is then validated by extensive experiments on benchmark datasets. Moreover, POP-Norm can have other forms that are much different from BatchNorm, and the principle of POP-Norm can be also applied to instance normalization and group normalization.

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

APPENDIX

A. POP-NORM FOR CONVOLUTION

When the layer is a convolutional layer, the matrix equivalence of a sample with respect to the learning parameter $w_k$ is

$$
x_{i_k} = \begin{bmatrix}
0 & 0 & \dots & 0 & x_{i_k}^{[1,1]} & \dots & 0 & \dots & 0 & x_{i_k}^{[d,1]} \\
0 & 0 & \dots & x_{i_k}^{[1,1]} & x_{i_k}^{[1,2]} & \dots & 0 & \dots & x_{i_k}^{[d,1]} & x_{i_k}^{[d,2]} \\
\dots & \dots & \dots & \dots & \dots & \dots & \dots & \dots & \dots & \dots \\
x_{i_k}^{[1,1]} & x_{i_k}^{[1,2]} & \dots & x_{i_k}^{[1,r-1]} & x_{i_k}^{[1,r]} & \dots & x_{i_k}^{[d,1]} & \dots & x_{i_k}^{[d,r-1]} & x_{i_k}^{[d,r]} \\
x_{i_k}^{[1,2]} & x_{i_k}^{[1,3]} & \dots & x_{i_k}^{[1,r]} & x_{i_k}^{[1,r+1]} & \dots & x_{i_k}^{[1,2]} & \dots & x_{i_k}^{[d,r]} & x_{i_k}^{[d,r+1]} \\
\dots & \dots & \dots & \dots & \dots & \dots & \dots & \dots & \dots & \dots \\
x_{i_k}^{[1,s-1]} & x_{i_k}^{[1,s]} & \dots & 0 & 0 & \dots & x_{i_k}^{[1,s-1]} & \dots & 0 & 0 \\
x_{i_k}^{[1,s]} & 0 & \dots & 0 & 0 & \dots & x_{i_k}^{[d,s]} & \dots & 0 & 0
\end{bmatrix},
\tag{17}
$$

where the number of channels is $d$, the number of spatial dimensions is $s$, the size of kernel is $r$. To make the expression legible, we present the matrix equivalence of input with one-dimensional spatial locations. Actually, when $s = 1$ and $r = 1$, the convolution layer is degraded to a fully-connected layer.

Theorem 2 and Theorem 3 require each row vector of $x_{i_k}$ in a convolutional should be considered as a atom that is like a sample in a fully-connected layer, so POP-Norm for a convolution layer is

$$
\bar{x}_{i_k}^{[j,q]} = x_{i_k}^{[j]} - \eta^{[j]} \cdot \frac{1}{ns} \sum_{i=1}^{n} \sum_{q=1}^{s} x_{i_k}^{[j,q]}, \qquad \text{(Mean Removal)} \tag{18}
$$

$$
\hat{x}_{i_k}^{[j,q]} = \frac{\bar{x}_{i_k}^{[j]}}{\rho^{[j]} \cdot \sqrt{\frac{1}{ns} \sum_{i=1}^{n} \sum_{i=1}^{s} \left( \bar{x}_{i_k}^{[j,q]} \right)^2}}, \qquad \text{(Division by sclaed } l_2 \text{ Norm)} \tag{19}
$$

B. PROOF OF THEOREM 1

**Proof 1** *(1) At the $k$-th iteration, we have*

$$
F(w_{k+1}) \le F(w_k) + \langle \nabla F(w_k), w_{k+1} - w_k \rangle + \frac{L}{2} \| w_{k+1} - w_k \|^2. \tag{20}
$$

*We know $w_{k+1} = w_k - \alpha_{k+1} \nabla f_{i_k}(w_k)$ in Eq. (2), so Eq. (20) can be formulated as*

$$
F(w_{k+1}) \le F(w_k) - \alpha_{k+1} \langle \nabla F(w_k), \nabla f_{i_k}(w_k) \rangle + \frac{L \alpha_{k+1}^2}{2} \| \nabla f_{i_k}(w_k) \|^2. \tag{21}
$$

*Since at each iteration the sample or mini-batch is randomly chose from the full set, $\nabla f_{i_k}(w_k)$ is an unbias estimate of $\nabla F(w)$, i.e., $\mathbb{E}[\nabla f_{i_k}(w)] = \nabla F(w)$. Hence taking expectation on the $k$-th sample or mini-batch, we have*

$$
\mathbb{E}[F(w_{k+1})] \le F(w_k) - \alpha_{k+1} \| \nabla F(w) \|^2 + \frac{L \alpha_{k+1}^2}{2} \mathbb{E}[\| \nabla f_{i_k}(w_k) \|^2]. \tag{22}
$$

*Rearranging terms yields*

$$
\alpha_{k+1} \| \nabla F(w) \|^2 \le F(w_k) - \mathbb{E}[F(w_{k+1})] + \frac{L \alpha_{k+1}^2}{2} \mathbb{E}[\| \nabla f_{i_k}(w_k) \|^2], \tag{23}
$$

*Taking the total expectation and summing over $0$ to $K$, we obtain*

$$
\sum_{k=0}^{K} \alpha_{k+1} \mathbb{E}[\| \nabla F(w_k) \|^2] \le \mathbb{E}[F(w_0)] - \mathbb{E}[F(W_{K+1})] + \frac{L}{2} \sum_{k=0}^{K} \alpha_{k+1}^2 \mathbb{E}[\| \nabla f_{i_k}(w_k) \|^2]. \tag{24}
$$

*Dividing both sides of Eq. (24) with $\sum_{k=0}^{K} \alpha_{k+1}$ and combining $F_{inf} \leq F(w_k)$, we have*

$$\frac{\sum_{k=0}^{K} \alpha_{k+1} \mathbb{E}[\|\nabla F(w_k)\|^2]}{\sum_{k=0}^{K} \alpha_{k+1}} \leq \frac{\mathbb{E}[F(w_0)] - F_{inf}}{\sum_{k=0}^{K} \alpha_{k+1}} + \frac{\frac{L}{2} \sum_{k=0}^{K} \alpha_{k+1}^2 \mathbb{E}[\|\nabla f_{i_k}(w_k)\|^2]}{\sum_{k=0}^{K} \alpha_{k+1}}. \tag{25}$$

*Since $\nabla F(w)$ is bounded, $\nabla f_{i_k}(w)$ will be bounded, and then $\mathbb{E}[\|\nabla f_{i_k}(w_k)\|^2]$ should be also bounded. It is known $\sum_{k=1}^{\infty} \alpha_k = \infty$ and $\sum_{k=1}^{\infty} \alpha_k^2 < \infty$, so when $K$ tends to infinity, we abtain*

$$\frac{\sum_{k=0}^{K} \alpha_{k+1} \mathbb{E}[\|\nabla F(w_k)\|^2]}{\sum_{k=0}^{K} \alpha_{k+1}} \xrightarrow{K \to \infty} 0. \tag{26}$$

*It is known that $\sum_{k=1}^{\infty} \alpha_k = \infty$, hence for any $\epsilon > 0$, there should exists a number $N \in \mathbb{N}$, so only sparse numbers $k > N$ satisfies that $\mathbb{E}[\|\nabla F(w_k)\|^2] > \epsilon$, otherwise when $K \to \infty$, $\frac{\sum_{k=0}^{K} \alpha_{k+1} \mathbb{E}[\|\nabla F(w_k)\|^2]}{\sum_{k=0}^{K} \alpha_{k+1}} \to \infty$ that contradicts Eq. (26). Therefore almost everywhere*

$$\mathbb{E}[\|\nabla F(w_k)\|^2] \xrightarrow{K \to \infty} 0. \tag{27}$$

*And then for any $\zeta > 0$,*

$$\mathbb{P}[\|\nabla F(w_k)\|] > \zeta] = \mathbb{P}[\|\nabla F(w_k)\|^2] > \zeta^2] \leq \frac{\mathbb{E}[\|\nabla F(w_k)\|^2]}{\zeta^2} \xrightarrow{K \to \infty} 0 \tag{28}$$

*where the inequality holds due to Markov's Inequality, which means $\|\nabla F(w_K)\| \xrightarrow{K \to \infty} 0$ in probability.*

*(2) Since $\frac{1}{\alpha_k} \geq L$ and Eq. (22) holds, we have*

$$\mathbb{E}[F(w_{k+1})] \leq F(w_k) - \alpha_{k+1} \|\nabla F(w)\|^2 + \frac{\alpha_{k+1}}{2} \mathbb{E}[\|\nabla f_{i_k}(w_k)\|^2]. \tag{29}$$

*It is known $\mathbb{E}[\|\nabla f_{i_k}(w_k)\|^2] = \mathbb{E}[\|\nabla f_{i_k}(w_k) - F(w)\|^2] + \|F(w)\|^2 = \mathbb{V}[\|\nabla f_{i_k}(w_k)\|] + \|F(w)\|^2$, so Eq. (29) can be reformulated as:*

$$\frac{\alpha_{k+1}}{2} \|\nabla F(w)\|^2 \leq F(w_k) - \mathbb{E}[F(w_{k+1})] + \frac{L\alpha_{k+1}^2}{2} \mathbb{V}[\|\nabla f_{i_k}(w_k)\|] \tag{30}$$

*Taking the total expectation and summing over $0$ to $K$, we have*

$$\frac{1}{2} \sum_{k=0}^{K} \alpha_{k+1} \mathbb{E}[\|\nabla F(w_k)\|^2] \leq \mathbb{E}[F(w_0)] - \mathbb{E}[F(W_{K+1})] + \sum_{k=0}^{K} \frac{\alpha_{k+1}}{2} \mathbb{V}[\|\nabla f_{i_k}(w_k)\|]. \tag{31}$$

*Dividing both sides of Eq. (31) with $\frac{1}{2} \sum_{k=0}^{K} \alpha_{k+1}$ and combining $F_{inf} \leq F(w_k)$, we have*

$$\frac{\sum_{k=0}^{K} \alpha_{k+1} \mathbb{E}[\|\nabla F(w_k)\|^2]}{\sum_{k=0}^{K} \alpha_{k+1}} \leq \frac{\mathbb{E}[F(w_0)] - F_{inf}}{\frac{1}{2} \sum_{k=0}^{K} \alpha_{k+1}} + \frac{\sum_{k=0}^{K} \alpha_{k+1} \mathbb{V}[\|\nabla f_{i_k}(w_k)\|]}{\sum_{k=0}^{K} \alpha_{k+1}}, \tag{32}$$

*It is known $\sum_{k=1}^{\infty} \alpha_k = \infty$, hence when $K \to \infty$ , we have $\frac{\mathbb{E}[F(w_0)] - F_{inf}}{\frac{1}{2} \sum_{k=0}^{K} \alpha_{k+1}} \to 0$. We know $\mathbb{E}[\|\nabla F(w_k)\|^2] \geq \inf_k \left( \mathbb{E}[\|\nabla F(w_k)\|^2] \right)$ and $\mathbb{V}[\|\nabla f_{i_k}(w_k)\|] \leq \sup_k \left( \mathbb{V}[\|\nabla f_{i_k}(w_k)\|] \right)$, and then we obtain*

$$\inf_k \left( \mathbb{E}[\|\nabla F(w_k)\|^2] \right) \leq \sup_k \left( \mathbb{V}[\|\nabla f_{i_k}(w_k)\|] \right), \tag{33}$$

*and finally we arrive at the desired result.* $\square$

## C. PROOF OF THEOREM 2

**Proof 2** *(1). From Eq. (8), we know $f_{i_k}(w_k; x_{i_k}) = g\left((x_{i_k})^t w_k + b_k\right)$, and then*

$$\nabla f_{i_k}(w_k; x_{i_k}) = x_{i_k} \nabla g\left((x_{i_k})^t w_k + b_k\right), \tag{34}$$

*hence*

$$
\begin{aligned}
\mathbb{V}\left[\|\nabla f_{i_k}(w_k; x_{i_k})\|\right] =& \mathbb{E}\left[\|\nabla f_{i_k}(w_k; x_{i_k})\|^2\right] - \left(\mathbb{E}\left[\|\nabla f_{i_k}(w_k; x_{i_k})\|\right]\right)^2 \\
\leq & \mathbb{E}\left[\|\nabla f_{i_k}(w_k; x_{i_k})\|^2\right] \\
=& \mathbb{E}\left[\|x_{i_k} \nabla g\left((x_{i_k})^t w_k + b_k\right)\|^2\right] \\
\leq & \mathbb{E}\left[\|x_{i_k}\|^2 \|\nabla g\left((x_{i_k})^t w_k + b_k\right)\|^2\right] \\
\leq & M_1 \mathbb{E}[\|x_{i_k}\|^2],
\end{aligned} \tag{35}
$$

*where the second inequality holds due to the fact $\|AB\|_2 \leq \|A\|_2 \|B\|_2$, and the third inequality follows $\|\nabla g(z)\|_2 \leq M_1 < \infty$.*

*Similarly, we conclude that*

$$\mathbb{V}\left[\|\nabla f_{i_k}(w_k; \bar{x}_{i_k})\|\right] \leq M_1 \mathbb{E}[\|\bar{x}_{i_k}\|^2] \tag{36}$$

*Recalling Eq. (9), we have*

$$
\begin{aligned}
\mathbb{E}[\|\bar{x}_{i_k}\|^2] = \mathbb{E}\left[\sum_{j=1}^d \left\|\bar{x}_{i_k}^{[j]}\right\|^2\right] =& \sum_{j=1}^d \mathbb{E}\left[\left\|\bar{x}_{i_k}^{[j]}\right\|^2\right] \\
=& \sum_{j=1}^d \mathbb{E}\left[\left\|x_{i_k}^{[j]} - \frac{\eta^{[j]}}{n}\sum_{i=1}^n x_{i_k}^{[j]}\right\|^2\right] \\
=& \sum_{j=1}^d \mathbb{E}\left[\left\|x_{i_k}^{[j]} - \eta^{[j]}\mathbb{E}\left[x_{i_k}^{[j]}\right]\right\|^2\right] \\
=& \sum_{j=1}^d \left(\mathbb{E}\left[\left\|x_{i_k}^{[j]}\right\|^2\right] - \eta^{[j]}(2 - \eta^{[j]})\left(\mathbb{E}\left[\left\|x_{i_k}^{[j]}\right\|\right]\right)^2\right) \\
=& \mathbb{E}[\|x_{i_k}\|^2] - \sum_{j=1}^d \eta^{[j]}(2 - \eta^{[j]})\left(\mathbb{E}\left[\left\|x_{i_k}^{[j]}\right\|\right]\right)^2,
\end{aligned} \tag{37}
$$

*From $0 \leq \eta^{[j]} \leq 2$ in Eq. (9), we know $0 \leq \eta^{[j]}(2 - \eta^{[j]}) \leq 1$, and then*

$$1 - \frac{(\mathbb{E}[\|x_{i_k}\|])^2}{\mathbb{E}[\|x_{i_k}\|^2]} \leq \frac{\mathbb{E}[\|\bar{x}_{i_k}\|^2]}{\mathbb{E}[\|x_{i_k}\|^2]} \leq 1 \tag{38}$$

*Hence we arrive the conclusion that mean removal in Eq. (9) will make the upper bound of of gradient variance $\mathbb{V}[\nabla f_{i_k}(w_k; \bar{x}_{i_k})]$ is lower than that of $\mathbb{V}[\nabla f_{i_k}(w_k; x_{i_k})]$.*

*(2). If Eq. (3) holds, the following inequality can be obtained due to Lagrange Mean Theorem, i.e.,*

$$L_{inf} = \|\nabla^2 F\left(w_k; \{x_{i_k}\}_{i=1}^n\right)\|_2, \tag{39}$$

*where $L_{inf}$ is the minimal gradient Lipschitz constant .*

*Following Eq. (8), we know $F\left(w_k; \{\bar{x}_{i_k}\}_{i=1}^n\right) = G\left(X_k^t w_k + B_k\right)$, and then the Hessian function of $F(w_k)$ is*

$$\nabla^2 F\left(w_k; \{x_{i_k}\}_{i=1}^n\right) = X_k \nabla^2 G\left(X_k^t w_k + B_k\right) X_k^t. \tag{40}$$

*And then we obtain*

$$
\begin{aligned}
\left\|\nabla^2 F\left(w_k ;\{x_{i_k}\}_{i=1}^n\right)\right\|_2 &= \left\|X_k \nabla^2 G\left(X_k^t w_k + B_k\right) X_k^t\right\|_2 \\
&= \left\|\nabla^2 G\left(X_k^t w_k + B_k\right) X_k^t X_k\right\|_2 \\
&\leq \left\|\nabla^2 G\left(X_k^t w_k + B_k\right)\right\|_2 \left\|X_k^t X_k\right\|_2 \\
&= \left\|\nabla^2 G\left(X_k^t w_k + B_k\right)\right\|_2 \left\|X_k X_k^t\right\|_2 \\
&\leq M_2 \left\|X_k X_k^t\right\|_2 \\
&\leq M_2 d \max_j \left(H_k^{[j,j]}\right),
\end{aligned}
\tag{41}
$$

*where $d$ is the dimension of each sample $x_{i_k}$ and $H_k^{[j,j]}$ are the $j$-th diagonal element of $(X_k X_k^t)$; the second and fourth equalities hold due to the fact $\|AB\|_2 = \|BA\|_2$ ; the third inequality follows $\|AB\|_2 \leq \|A\|_2 \|B\|_2$; the fifth inequality is owing to $\|\nabla^2 G(z)\|_2 \leq M_2$, and the last inequality is due to the fact $\|A\|_2 \leq d \max_j(A^{[j,j]})$ where $A \in \mathbb{R}^{d \times d}$ is positive definite and $A^{[j,j]}$ is the $j$-th diagonal element.*

*Similarly, we can derive that*

$$
\left\|\nabla^2 F\left(w_k ;\{\bar{x}_{i_k}\}_{i=1}^n\right)\right\|_2 \leq M_2 d \max_j \left(\bar{H}_k^{[j,j]}\right),
\tag{42}
$$

*where $\bar{H}_k^{[j,j]}$ is $j$-th diagonal element of $(\bar{X}_k \bar{X}_k^t)$ in which $\bar{X}_k$ is constructed by following $X_k$ in Eq. (8).*

*According to the definition of $\bar{X}_k$ in Eq. (9), $\bar{H}_k^{[j,j]}$ can be reformulated as*

$$
\begin{aligned}
\bar{H}_k^{[j,j]} &= \sum_{i=1}^n \left(\bar{x}_{i_k}^{[j]}\right)^2 = \sum_{i=1}^n \left(x_{i_k}^{[j]} - \frac{\eta^{[j]}}{n} \sum_{i=1}^n x_{i_k}^{[j]}\right)^2 \\
&= \sum_{i=1}^n \left(x_{i_k}^{[j]}\right)^2 - \frac{\eta^{[j]}(2 - \eta^{[j]})}{n} \left(\sum_{i=1}^n x_{i_k}^{[j]}\right)^2 \\
&= H_k^{[j,j]} - \frac{\eta^{[j]}(2 - \eta^{[j]})}{n} \left(\sum_{i=1}^n x_{i_k}^{[j]}\right)^2
\end{aligned}
\tag{43}
$$

*From $0 \leq \eta^{[j]} \leq 2$ in Eq. (9), we know $0 \leq \eta^{[j]}(2 - \eta^{[j]}) \leq 1$, and then*

$$
1 - \left(\sum_{i=1}^n x_{i_k}^{[j]}\right)^2 \leq \frac{\bar{H}_k^{[j,j]}}{H_k^{[j,j]}} \leq 1
\tag{44}
$$

*Therefore the upper bound of the gradient Lipschitz constant $L_{inf}$ of $F\left(w_k ;\{\bar{x}_{i_k}\}_{i=1}^n\right)$ is lower that of $F\left(w_k ;\{x_{i_k}\}_{i=1}^n\right)$.$\square$*

## D. PROOF OF THEOREM 3

**Proof 3** *(1)Similar to Eq.(34 - 35), it can be also concluded that*

$$
\mathbb{V}\left[\nabla f_{i_k}(w_k ; \hat{x}_{i_k})\right] \leq M_1 \mathbb{E}[\|\hat{x}_{i_k}\|^2]
\tag{45}
$$

*According to Eq.(10), we have*

$$
\begin{aligned}
\mathbb{E}[\|\hat{x}_{i_k}\|^2] &= \sum_{j=1}^{d} \mathbb{E}\left[\left\|\hat{x}_{i_k}^{[j]}\right\|^2\right] \\
&= \sum_{j=1}^{d} \mathbb{E}\left[\left\|\frac{\bar{x}_{i_k}^{[j]}}{\rho^{[j]} \cdot \sqrt{\frac{1}{n}\sum_{i=1}^{n}\left(\bar{x}_{i_k}^{[j]}\right)^2}}\right\|^2\right] \\
&= \frac{1}{\frac{(\rho^{[j]})^2}{n}\sum_{i=1}^{n}\left(\bar{x}_{i_k}^{(j)}\right)^2}\mathbb{E}[\|\bar{x}_{i_k}\|^2]
\end{aligned}
\tag{46}
$$

*Hence,*

$$
\frac{\mathbb{E}[\|\hat{x}_{i_k}\|^2]}{\mathbb{E}[\|\bar{x}_{i_k}\|^2]} = \frac{1}{(\rho^{[j]})^2}\frac{1}{\frac{1}{n}\sum_{i=1}^{n}\left(\bar{x}_{i_k}^{(j)}\right)^2} \leq 1
\tag{47}
$$

*where the inequality holds since $\frac{(\rho^{[j]})^2}{n}\sum_{i=1}^{n}\left(\bar{x}_{i_k}^{(j)}\right)^2 \geq 1$.*

*Therefore the upper bound of the gradient variance $\mathbb{V}[\nabla f_{i_k}(w_k; \hat{x}_{i_k})]$ will be lowered than that of $\mathbb{V}[\nabla f_{i_k}(w_k; \bar{x}_{i_k})]$.*

*(2). Similar to Eq. (35 - 40), we conclude*

$$
\|\nabla^2 F(w_k; \{\hat{x}_{i_k}\}_{i=1}^{n})\|_2 \leq M_2 d \max_{j}\left(\hat{H}_{k}^{[j,j]}\right),
\tag{48}
$$

*where $\hat{H}_{k}^{[j,j]}$ is $j$-th diagonal element of $(\hat{X}_k \hat{X}_k^t)$ in which $\hat{X}_k$ is constructed by following $X_k$ in Eq. (8).*

*According to Eq.(10), we have*

$$
\hat{H}_{k}^{[j,j]} = \sum_{i=1}^{n}\left(\hat{x}_{i_k}^{[j]}\right)^2 = \sum_{i=1}^{n}\left(\frac{\bar{x}_{i_k}^{[j]}}{\rho^{[j]}\cdot\sqrt{\frac{1}{n}\sum_{i=1}^{n}\left(\bar{x}_{i_k}^{[j]}\right)^2}}\right)^2
\tag{49}
$$

*Hence,*

$$
\frac{\hat{H}_{k}^{[j,j]}}{\bar{H}_{k}^{[j,j]}} = \frac{1}{(\rho^{[j]})^2}\frac{1}{\frac{1}{n}\sum_{i=1}^{n}\left(\bar{x}_{i_k}^{(j)}\right)^2} \leq 1,
\tag{50}
$$

*where the inequality holds since $\frac{(\rho^{[j]})^2}{n}\sum_{i=1}^{n}\left(\bar{x}_{i_k}^{(j)}\right)^2 \geq 1$.*

*Hence the upper bound of the minimal gradient Lipschitz constant of $F(w_k; \{\hat{x}_{i_k}\}_{i=1}^{n})$ will be further lowered than that of $F(w_k; \{\bar{x}_{i_k}\}_{i=1}^{n})$.*

*(3). If $F(w_k; \{\bar{x}_{i_k}\}_{i=1}^{n})$ is locally strongly convex, the condition num $G(z)$ is also strong convex, and then*

$$
\frac{\lambda_{min}(G(z))}{\lambda_{max}(G(z))} > M > 0,
\tag{51}
$$

*where $\lambda_{\max}(\cdot)$ is the maximal eigenvalue and $\lambda_{\min}(\cdot)$ is the minimal eigenvalue.*

the local space the condition number of $\nabla^2 F_{(c)}(w_k; \{\bar{x}_{i_k}\}_{i=1}^n)$ is

$$
\begin{aligned}
\text{cond}(\nabla^2 F\left(w_k; \{\bar{x}_{i_k}\}_{i=1}^n\right)) =& \|\nabla^2 F\left(w_k; \{\bar{x}_{i_k}\}_{i=1}^n\right)\|_2 \| \left(\nabla^2 F\left(w_k; \{\bar{x}_{i_k}\}_{i=1}^n\right)\right)^{-1} \|_2 \\
=& \|\bar{X}_k \nabla^2 G\left(\bar{X}_k^t w_k + B_k\right) \bar{X}_k^t\|_2 \| \left(\bar{X}_k \nabla^2 G\left(\bar{X}_k^t w_k + B_k\right) \bar{X}_k^t\right)^{-1}\|_2 \\
=& \|\nabla^2 G\left(\bar{X}_k^t w_k + B_k\right) \bar{X}_k^t \bar{X}_k\|_2 \|\nabla^2 \left(G\left(\bar{X}_k^t w_k + B_k\right)\right)^{-1} (\bar{X}_k^t \bar{X}_k)^{-1}\|_2 \\
\geq& \frac{\lambda_{\min}(G)}{\lambda_{\max}(G)} \cdot \frac{\lambda_{\max}(\bar{X}_k \bar{X}_k^t)}{\lambda_{\min}(\bar{X}_k \bar{X}_k^t)} \\
\geq& M \frac{\lambda_{\max}(\bar{X}_k \bar{X}_k^t)}{\lambda_{\min}(\bar{X}_k \bar{X}_k^t)} \\
\geq& M \frac{\max_j \left(\bar{H}_k^{[j,j]}\right)}{\min_j \left(\bar{H}_k^{[j,j]}\right)},
\end{aligned}
$$
(52)

where $\bar{H}_k^{[j,j]}$ is the j-th diagonal element of $(\bar{X}_k \bar{X}_k^t)$; The third equality follows $\|AB\|_2 = \|BA\|_2$; the forth inequality holds due to the fact $\|AB\|_2 \geq \frac{1}{d}\text{Tr}(AB) \geq \lambda_{\min}(A)\lambda_{\max}(B)$ if A and B are positive definite.

Similarly, we have

$$
\text{cond}(\nabla^2 F\left(w_k; \{\hat{x}_{i_k}\}_{i=1}^n\right)) \geq M \frac{\max_j \left(\hat{H}_k^{[j,j]}\right)}{\min_j \left(\hat{H}_k^{[j,j]}\right)}
$$
(53)

Recalling Eq. (10), we obtain

$$
\hat{H}_k^{[j,j]} = \sum_{i=1}^n \left(\hat{x}_{i_k}^{[j]}\right)^2 = \sum_{i=1}^n \left(\frac{\bar{x}_{i_k}^{[j]}}{\rho^{[j]} \cdot \sqrt{\frac{1}{n}\sum_{i=1}^n \left(\bar{x}_{i_k}^{[j]}\right)^2}}\right)^2 = (\rho^{[j]})^2
$$
(54)

It is known $\frac{\max_j((\rho^{[j]})^2)}{(\min_j(\rho^{[j]})^2)} \leq \frac{\max_j \left(\sum_{i=1}^n (\bar{x}_{i_k}^{[j]})^2\right)}{\min_j \left(\sum_{i=1}^n (\bar{x}_{i_k}^{[j]})^2\right)} = \frac{\max_j \left(\bar{H}_k^{[j,j]}\right)}{\min_j \left(\bar{H}_k^{[j,j]}\right)}$, hence we have

$$
\frac{\max_j \left(\hat{H}_k^{[j,j]}\right)}{\min_j \left(\hat{H}_k^{[j,j]}\right)} \leq \frac{\max_j \left(\bar{H}_k^{[j,j]}\right)}{\min_j \left(\bar{H}_k^{[j,j]}\right)}.
$$
(55)

Finally we arrive the conclusion that the lower bound of conditional number of $\nabla^2 F\left(w_k; \{\hat{x}_{i_k}\}_{i=1}^n\right)$ will be lowered than that of $\nabla^2 F\left(w_k; \{\bar{x}_{i_k}\}_{i=1}^n\right)$ in the local space. $\square$

