# OpenReview forum: "POP-Norm: A Theoretically Justified and More Accelerated Normalization Approach"
_ICLR.cc/2020/Conference — Reject_

### Official Review · AnonReviewer2 · 2019-10-15
**Official Blind Review #2**

**Rating:** 3

**Review:**

This paper proposes Pre-OPeration Normalization (POP-norm) as a theoretically grounded modification of batch normalization (Batch-norm). The authors provide an analysis of stochastic gradient descent (SGD) showing that the convergence can be accelerated by lowering the gradient Lipschitz constant and gradient variances. Based on this observation, the authors show that by applying POP-norm which is quite similar to the Batch-norm applied before the activation functions, the gradient Lipschitz constants and the gradient variances can be lowered.

I must first confess that I’m not an expert in the relevant area. I’m don’t know why the reviewing system assigned this paper to me, but I will try my best to judge the value of the paper.

 Having said that, I don’t have enough background knowledge to evaluate the novelty of the paper. For instance, the Theorem 1 is stated as if it was originally introduced in this paper, without any citation. But I doubt that because I could easily find several results relating the convergence rate of SGD to the gradient Lipschitz constant. Are there any papers or standard textbooks to be noted? If so, please cite them properly.

The theorem 2 and 3 are based on the full-dataset statistics, but because they are infeasible in practice, the authors propose to use the batch statistics. How may this affect the theory? Intuitively, one could easily imagine that the smaller batch-size would lead to higher variances for batch statistics estimators, and thus impeding the regularization effect of the normalization. Also, the theory just states that the upper bound of (Lipschitz constant, gradient variance) after normalization will be lower than the lower bound without normalization, so it is not clear how much the normalization would reduce them. If possible please state the detailed results in the main text.

The differences between POP-norm and Batch-norm are, 1) whether the normalization is applied before or after the operations, and 2) minor difference in the normalization procedure, such as the additional scaling parameter kappa. The experiments provide results to demonstrate the effectiveness of 1), but not clear about 2). It would be interesting to compare POP-norm to the Batch-norm applied before the operations to see the net effect of 2). The very idea of applying Batch-norm before the convolution operation has already been demonstrated and widely used in the case of ResNet with ReLU activation functions, so if the contribution from 2) is insignificant, I think one may ask the novelty of the proposed normalization scheme.

Overall, despite the lack of background knowledge, I find this paper to be borderline, but more close to the reject, because of some unclear points. I would like to hear back from the authors and other reviewers.

I don’t understand why the matrix B in eq.8 gathering the biases has additional indices 1_k, …, m_k. Shouldn’t it be the repetition of b_k for m times?

**Experience Assessment:**

I do not know much about this area.

**Review Assessment: Checking Correctness Of Derivations And Theory:**

I assessed the sensibility of the derivations and theory.

**Review Assessment: Checking Correctness Of Experiments:**

I carefully checked the experiments.

**Review Assessment: Thoroughness In Paper Reading:**

I read the paper at least twice and used my best judgement in assessing the paper.

---

### Official Review · AnonReviewer3 · 2019-10-21
**Official Blind Review #3**

**Rating:** 3

**Review:**

This paper proposed a new normalization method for accelerating the training of deep neural networks. Specifically, the proposed method put the normalization step before the linear transformation step. Then, the authors show that the normalization step can reduce the variance of gradients. The experimental results show some improvement.

1. The proof of Theorem 1 is the regular convergence analysis of SGD. No new contributions.

2. If my understanding is right, Theorem 2 and 3 actually can also show that other normalization techniques have these effects. So, what's the difference between POP-Norm and other normalization techniques in reducing gradient variance and  Lipschitz  constant?

3. This paper claims that the effect of batch normalization will be destroyed by Tanh and Sigmoid activation function. How about POP-Norm? It's better to show the performance of POP-Norm for the network with Tanh or Sigmoid.

4. How about its performance for large networks, such as ResNet-50?

**Experience Assessment:**

I have published one or two papers in this area.

**Review Assessment: Checking Correctness Of Derivations And Theory:**

I assessed the sensibility of the derivations and theory.

**Review Assessment: Checking Correctness Of Experiments:**

I carefully checked the experiments.

**Review Assessment: Thoroughness In Paper Reading:**

I read the paper thoroughly.

---

### Official Review · AnonReviewer1 · 2019-10-21
**Official Blind Review #1**

**Rating:** 1

**Review:**

Paper summary: This paper proposes an alternative approach to batch normalization called POP-Norm. The authors motivate their approach theoretically and suggest that this also helps explain BatchNorm’s success due to the similarity between the two approaches. They then empirically examine their method on CIFAR10, CIFAR100 and ImageNet classification.

Comments: I find the paper hard to read and and plagued with major technical flaws. The paper also shows unfamiliarity with standard literature on optimization. Some of my major concerns are discussed below:

A. Theoretical Results (Section 2):

These results (the connections between convergence and gradient Lipshictzness (beta-smoothness) and gradient variance) are extremely well-known (an example reference is Section 3 in arXiv:1909.03550). In fact, the effects of gradient variance on convergence in optimization have prompted a whole sub-field on variance reduced gradient descent methods such as SVRG. Moreover, prior work by Santurkar et al. identifies precisely the same effect (better beta-smoothness) with BatchNorm as one of the factors responsible for its success.

B. Proposed approach:

The authors claim that one of the key differences between their approach and BatchNorm is dividing by the l2 norm instead of the gradient variance. However, it seems that they take the l2 norm of the random variable after mean subtraction not before (Equations 10 and 12). This is actually the same as the variance of the variable before mean subtraction, which makes this part of their approach identical to BatchNorm. Then the only difference in this aspect of their approach is the way the learned scale and shift is introduced, which seems somewhat arbitrary.

Even more concerning are the claims in Theorems 2 and 3 which justify why mean subtraction and dividing by the l2 norm (variance) help reduce gradient Lipschitz constant/variance. The authors are comparing two upper (or two lower) bounds which is insufficient to draw any meaningful conclusions. In particular, given two random variables X1 and X2, one cannot claim that X1 < X2 based on (arbitrary) upper bounds; especially since there is no evidence that these bounds are tight.

Moreover, all the theoretical analysis in this Section is performed for a single layer of the network assuming other layers are fixed. Specifically, as per their proofs, the gradient of the weights at a particular layer is not dependent by any of the previous/layer layers, thus completely ignoring the effects that come from simultaneously updating all the layers. Thus, the theoretical analysis in this Section does not really tell us about the effects of BatchNorm in a deep network. Their setting is actually identical to studying the effects of input normalization on a single network layer. In this setting, it is well-known that input normalization improves the conditioning of the problem and thus helps optimization ([LeCun et al., 1998 and Wiesler et al., 2011])---in fact, this was actually one of the motivations of the original BatchNorm paper.

C. Experimental Results:

Based on this, it seems to me that the proposed approach is essentially the same as BatchNormalization except for (1) the positioning of the layer (before the weights instead of after), (2) the exact way the learned scale and shift is done and (3) a hyperparameter kappa that is used to scale down the variance of all the layers and hence essentially scales down the learning rate by a constant.

From an experimental point of view, I find (3) particularly concerning because it just means that in their learning rate for the POP-Norm experiments are 1/kappa times those for the BatchNorm baseline. Since the authors do not really tune over hyperparameters (they use a fixed learning rate in all their experiments) when comparing the different approaches, it is not clear whether the chosen learning rate is optimal for BatchNorm. To perform a fair comparison of their approach to BatchNorm, the authors should:
1. Introduce the kappa scaling into BatchNorm layers and see the effect on performance.
2. Perform a grid search for both methods and report the best performance over hyperparameters.
3. Also include comparison to other SOTA normalization alternatives (including those that the authors mention in the introduction)

D. Other comments:
i. The discussion in Section 4 is hard to parse and I am not convinced of its correctness (I am not sure how the authors arrived at (16)).
ii. The paper writing is poor with several typos and spelling mistakes. In addition, the inline math makes the paper very hard to read at times.

Overall, I think there are significant concerns with the paper, both in terms of writing and the soundness/novelty of the technical results and experiments. Thus, I recommend rejection.

------------------------------------------
References:

LeCun, Y., Bottou, L., Orr, G., and Muller, K. Efficient backprop. In Orr, G. and K., Muller (eds.), Neural Networks: Tricks of the trade. Springer, 1998.

Wiesler, Simon and Ney, Hermann. A convergence analysis of log-linear training. In Shawe-Taylor, J., Zemel, R.S., Bartlett, P., Pereira, F.C.N., and Weinberger, K.Q. (eds.), Advances in Neural Information Processing Systems 24, pp. 657–665, Granada, Spain, December 2011.


**Experience Assessment:**

I have published one or two papers in this area.

**Review Assessment: Checking Correctness Of Derivations And Theory:**

I carefully checked the derivations and theory.

**Review Assessment: Checking Correctness Of Experiments:**

I carefully checked the experiments.

**Review Assessment: Thoroughness In Paper Reading:**

I read the paper thoroughly.

---

### Decision · Program_Chairs · 2019-12-19

**Decision:**

Reject

**Comment:**

The authors propose an alternative to batch norm, which they call POP-norm, and provide theoretical justification for POP-norm in nonconvex optimization on the basis of variance reduction. They then present empirical arguments.

One of the most cogent reviewers believed the theoretical results were known and the empirical arguments unconvincing because the method is similar to batch norm up to a change in learning rate and some minor differences.

Unfortunately, the reviewers did not engage with the author rebuttals at all. The authors seem to have addressed most points. However, if the reviewers are unwilling to engage, despite multiple emails, there's not much I can do, short of redoing the whole process from scratch. And I'll take the lack of engagement as lack of interest by the reviewers. Not being an expert in optimization myself, I'm not going to override the scores. I do know enough to know that there are standard bounds for both convex and nonconvex optimization that improve with decreased variance.